# PP1, PP2A and PP2B Interplay in the Regulation of Sperm Motility: Lessons from Protein Phosphatase Inhibitors

**DOI:** 10.3390/ijms232315235

**Published:** 2022-12-03

**Authors:** Ana F. Ferreira, Joana Santiago, Joana V. Silva, Pedro F. Oliveira, Margarida Fardilha

**Affiliations:** 1Laboratory of Signal Transduction, Institute for Biomedicine-iBiMED, Medical Sciences Department, University of Aveiro, 3810-193 Aveiro, Portugal; 2QOPNA & LAQV, Department of Chemistry, University of Aveiro, 3810-193 Aveiro, Portugal

**Keywords:** sperm motility, capacitation, protein phosphatase type 1, protein phosphatase type 2A, protein phosphatase type 2B, protein phosphatase inhibitors

## Abstract

Male fertility relies on the ability of spermatozoa to fertilize the egg in the female reproductive tract (FRT). Spermatozoa acquire activated motility during epididymal maturation; however, to be capable of fertilization, they must achieve hyperactivated motility in the FRT. Extensive research found that three protein phosphatases (PPs) are crucial to sperm motility regulation, the sperm-specific protein phosphatase type 1 (PP1) isoform gamma 2 (PP1γ2), protein phosphatase type 2A (PP2A) and protein phosphatase type 2B (PP2B). Studies have reported that PP activity decreases during epididymal maturation, whereas protein kinase activity increases, which appears to be a requirement for motility acquisition. An interplay between these PPs has been extensively investigated; however, many specific interactions and some inconsistencies remain to be elucidated. The study of PPs significantly advanced following the identification of naturally occurring toxins, including calyculin A, okadaic acid, cyclosporin, endothall and deltamethrin, which are powerful and specific PP inhibitors. This review aims to overview the protein phosphorylation-dependent biochemical pathways underlying sperm motility acquisition and hyperactivation, followed by a discussion of the PP inhibitors that allowed advances in the current knowledge of these pathways. Since male infertility cases still attain alarming numbers, additional research on the topic is required, particularly using other PP inhibitors.

## 1. Introduction

Male fertility relies on the ability of spermatozoa to fertilize the egg in the female reproductive tract (FRT), which requires them to be capable of moving after ejaculation. When spermatogenesis is completed, spermatozoa leave the testis as morphologically complete but functionally immature cells. To acquire fertilization ability, the spermatozoon needs to undergo maturation in the epididymis and capacitation in the FRT, to acquire activated and hyperactivated motility, respectively [1,2]. The root of some of the most prevalent male infertility disorders, such as asthenozoospermia (poor sperm motility), is usually unknown due to a lack of understanding of the molecular pathways that regulate sperm function [3,4,5]. In fact, a high percentage of motile spermatozoa is a crucial component for fertility, since it is considered a strong predictive marker of fertility potential [5].

Since spermatozoa are virtually absent of gene expression and protein synthesis, post-translational modifications, particularly protein phosphorylation, should play a crucial role in regulating sperm motility acquisition [6,7] through an intricate interplay between protein kinases (PKs) and protein phosphatases (PPs) [2,4,7]. In particular, it is well known that Phosphoprotein Phosphatase 1 catalytic subunit gamma 2 (PP1γ2), a PP1 isoform present only in testes and sperm, is essential for sperm motility acquisition [8,9,10]. The study of PP’s role in cellular processes, including cytoskeleton organization, cell cycle control and apoptosis [11], critically advanced through exploring the properties of small toxins, which are powerful PP inhibitors. To date, only a small portion of the currently known PP inhibitors was reported to be tested in spermatozoa, with the purpose of PP inhibition, namely calyculin A (CA) okadaic acid (OA), cyclosporin (CsA), endothall (E) and deltamethrin (DEL) [12,13,14,15,16].

Extensive research on this topic allowed the unveiling of many specific interactions between PPs; however, a complete and updated interplay involving protein phosphatase type 1 gamma 2 (PP1γ2), protein phosphatase type 2A (PP2A) and protein phosphatase type 2B (PP2B) was not yet reviewed. Hence, this review comprises an overview of the phosphorylation-dependent biochemical mechanisms underlying motility acquisition and hyperactivation in spermatozoa, aiming to describe the PPs interplay, as well as a discussion of the PP inhibitors that allowed critical advances in their current knowledge.

## 2. Activated and Hyperactivated Sperm Motility

According to Paoli et al., sperm motility can be defined as a propagation of transverse waves along the flagellum in a proximal-distal direction, which produces an impulse that pushes the spermatozoon towards the female gamete [17]. This is achieved when spermatozoa own a morphologically complete flagellum, when they can produce energy to fuel the movement and present functional signaling pathways that efficiently regulate protein phosphorylation status [4,17].

Spermatozoa acquire motility across their passage along the epididymis, emerging with activated progressive motility after complete and successful maturation. The epididymis consists of three major regions, the caput which is the proximal end that succeeds the testes, followed by the corpus and the cauda, where sperm is stored until ejaculation [1,4]. Some authors consider an additional anatomical region in rodents, the initial segment, positioned between the testes and the caput region; however, its presence in other mammals was not clearly described [2]. Several studies confirm that spermatozoa enter the caput epididymis immotile and endure diverse morphologic, metabolic, and biochemical changes until they reach the caudal end progressively motile. These alterations include changes in intracellular concentrations of calcium (Ca^2+^) and cyclic adenosine monophosphate (cAMP), pH and phosphorylation status of critical amino acid residues [2,3,4,9,18,19]. The resulting activated motility is characterized by low-amplitude symmetrical tail movements that drive the spermatozoon in a straight line in non-viscous media, such as the seminal plasma [3,19].

Despite being progressively motile when ejaculated, spermatozoa are not capable of fertilization, they must acquire hyperactivated motility in the FRT, presumably in the fallopian tubes, through the capacitation process [3,5,20]. Capacitation is triggered by the unique environment of the FRT and causes spermatozoa to undergo a specific cascade of biochemical and physiological alterations, that ultimately allows them to reach and fertilize the oocyte [21,22,23,24]. Overall, capacitation involves the removal of the membrane cholesterol by albumin, which increases membrane permeability and hyperpolarization [25,26]. Consequently, there is a rise in sperm intracellular pH, associated with higher bicarbonate (HCO_3_^−^) concentration, which increases Ca^2+^ uptake [27,28,29]. This leads to the activation of the testis-specific soluble adenylyl cyclase (sAC), which produces the secondary messenger cAMP. This event activates protein kinase A (PKA) which subsequently initiates a protein tyrosine phosphorylation cascade [19,22,23,30]. Hyperactivated motility is characterized by asymmetric and high amplitude flagellar bends, allowing the spermatozoon to move along the dense mucus of the FRT in a circular or figure-eight trajectory [3,21]. In vitro hyperactivated motility induction requires Ca^2+^, HCO_3_^−^ and other metabolic substrates that mimic the female tract environment [23,31]. Capacitation and hyperactivation constitute distinct processes that can happen separately, since they are regulated by different biochemical players. However, to some extent, these mechanisms overlap, and the two processes are usually reported in association [3,24,32].

## 3. Protein Phosphatases and Their Role in Spermatozoa Function

The equilibrium of protein phosphorylation systems is essential to maintain cellular viability and function. Indeed, it is the most common post-translational type of modification in eukaryotes [33,34]. Since spermatozoa are virtually transcriptionally and translationally inactive, their specific functions are mediated mainly by protein phosphorylation [2,6,7]. Together with the extensively researched PKs, sperm-specific PPs are known to have critical roles in spermatozoa maturation, especially regarding sperm motility acquisition during epididymal transit (Figure 1a) and sperm hyperactivation in the FRT (Figure 1b) [2,9,19,35,36]. According to Smith et al., in bovine caput epididymis, immotile spermatozoa have higher phosphatase activity when compared to caudal motile sperm [8].

### 3.1. Protein Phosphatase Type 1 (PP1)

PP1 is the predominant PP identified in the spermatozoon [8,10]. Since the first description of its association with sperm motility in 1996 [9], PP1’s role in the signaling events involved in spermatozoa motility acquisition within the epididymis has been investigated [4,10,19]. The catalytic subunit of PP1 is highly conserved among eukaryotes; it consists of a single domain of ~30 kDa that binds to regulatory subunits. PP1 catalytic activity is due to the channel formed by three β-sheets of the β-sandwich, along with two metal ions (Manganese (Mn) and Iron (Fe)) coordinated by six amino acid residues (one Asn, three His, and two Asp) that form the active site [37,38]. In mammals, the catalytic subunit of PP1 presents four distinct isoforms—PP1α, PP1β, PP1γ1 and PP1γ2—encoded in three different genes (*Ppp1ca, Ppp1cb, Ppp1cc*). PP1γ1 and PP1γ2 result from alternative splicing of the same gene, *Ppp1cc*, being the first ubiquitously expressed, along with PP1α and PP1β, and the latest testis-enriched and sperm-specific isoform. The two isoforms’ amino acid sequences are almost identical, differing only at the C-termini [8,10,39,40]. The different isoforms of the catalytic subunit present distinct subcellular localization patterns and exist in the cell in association with regulatory subunits, the PP1 interacting proteins [PIPs; also known as regulatory interactors of protein phosphatase one (RIPPOs)] [11,41,42]. More than 800 PIPs were identified so far, which guide PP1 action within the cell, specify PP1 substrates and regulate PP1 activity [43,44].

#### Protein Phosphatase 1 Gamma 2 (PP1γ2)

PP1γ2 is the testis-enriched and sperm-specific PP1, which in mammals appears to be the principal isoform responsible for PP1 activity in spermatozoa [8,9,10]. It differs from the PP1γ1 on the C-terminal and is distributed throughout the flagellum, midpiece, and posterior region of the head of the spermatozoon [10,11]. Numerous studies showed that the decrease of PP1γ2 activity is associated with increased motility in the caudal epididymis, whereas immotile caput spermatozoa present high levels of PP1γ2 activity. During capacitation, the downregulation of this PP activity in spermatozoon was also evident [8,9,31,45]. Phosphatase activity inhibition in caput epididymis, by both PP1 inhibitors CA and OA, was also able to induce motility [8,9]. Additionally, *Ppp1cc* gene knockout in mice, which causes the loss of both PP1γ1 and PP1γ2, resulted in impaired spermatogenesis and subsequent male infertility [46,47]. Conditional knockout of only PP1γ2 resulted in the same phenotype, strongly suggesting that *Ppp1cc* knockout mice infertility is likely due to the loss of PP1γ2 [48]. Besides, PP1γ2 transgenic expression in *Ppp1cc* null mice was able to restore sperm function and fertility [49]. Hence, it is thought that PP1γ2 is responsible for the role of PP1 in motility acquisition in the epididymis, hyperactivation and acrosome reaction [4].

In sperm, PP1γ2 activity is mainly regulated by three specific inhibitors, protein phosphatase inhibitor 2 (PPP1R2, also known as I2), protein phosphatase 1 regulatory subunit 7 (PPP1R7, also known as SDS22) and protein phosphatase 1 regulatory subunit 11 (PPP1R11, also known as I3), whose association with PP1γ2 varies during epididymal sperm maturation [7,50,51]. In brief, PP1γ2 is solely bound to I3 in immotile spermatozoa of the caput epididymis, while in caudal spermatozoa it is bound to all three inhibitors. These alterations play an important role in motility development because they can modulate PP1γ2 activity [50]. In 2013, a PPP1R2 isoform was identified in human spermatozoa, the protein phosphatase inhibitor 2-like (PPP1R2P3), which appears to be present only in caudal spermatozoa, inhibiting PP1γ2 and therefore contributing to motility acquisition. Due to Thr73 being substituted by proline, PPP1R2P3 cannot be phosphorylated by glycogen synthase kinase 3 (GSK3) [52]. Furthermore, recently Schwartz et al. identified another possible PP1γ2 binding protein, which is CCDC181. Despite little being known about their interaction, the authors hypothesize that CCDC181 has a relevant role in generating and regulating flagellar and ciliary motility [53].

GSK3 (recently reviewed in Dey et al. 2019 [19]) was discovered to interact with PP1γ2 in the spermatozoon, playing an essential role in its activation [9,54]. Similar to PP1γ2, GSK3 presents six times more catalytic activity in immotile caput spermatozoa when compared to motile caudal sperm [54,55]. Currently, it is known that GSK3 regulates I2 binding to PP1γ2; however, it is only speculated that the other PP1γ2 inhibitors referred (SDS22 and I3) are also regulated by phosphorylation [9,50].

### 3.2. Protein Phosphatase Type 2A (PP2A)

PP2A activity was first documented in mammalian spermatozoa in 1996 by Vijayaraghavan et al. [9]. This Ser/Thr PP consists of a catalytic subunit (PP2A-C), which has two isoforms (α and β), a scaffolding subunit (PP2A-A, also with two isoforms) and a regulatory subunit (PP2A-B) [56,57]. The catalytic subunit of PP2A (36 kDa), which is one of the most conserved in eukaryotes, and the scaffolding subunit (65 kDa) form the enzyme core that can associate with different regulatory subunits giving rise tothe PP2A holoenzyme. The catalytic subunit consists of a typical α/β fold and contains two Mn ions at the enzyme’s active site [58,59]. It can be covalently modified by either Tyr phosphorylation or carboxymethylation [57,59].

It was reported that PP2A plays a role in both bovine and human sperm motility acquisition. Similar to PP1γ2, higher levels of this PP activity were identified in immotile caput spermatozoa, and downregulation of its activity was evident in both caudal motile and hyperactivated spermatozoa. In addition, inhibition of PP2A prevented motility initiation in caput epididymal spermatozoa but was able to stimulate it at the caudal sperm level. Changes in PP2A methylation also modify spermatozoa movement, as methylated PP2A is catalytically inactive in caudal spermatozoa [31,57,60]. GSK3 is a known target of PP2A, since its phosphorylation was increased following PP2A inhibition. Hence, PP2A is thought to be involved in sperm motility mainly by regulating GSK3 activity [57].

### 3.3. Phosphoprotein Phosphatase Type 2B (PP2B)

PP2B, also known as calcineurin, is a Ser/Thr PP regulated by Ca^2+^, which was also found in humans and other mammalian spermatozoa [35,61]. This PP is Ca^2+^/calmodulin (CAM)-dependent, meaning that it is inactive when it is not associated with Ca^2+^-CAM [61]. PP2B is composed of two subunits, catalytic and regulatory. There are three isoforms of the catalytic subunit (PPP3CA, PPP3CB, and PPP3CC), PPP3CC being the sperm-specific catalytic isoform. The regulatory subunit also presents two isoforms PPP3R1 and PPP3R2, the latest being present in spermatozoa. Absence of both sperm-specific isoforms due to gene knockout results in male infertility [35,36,62,63]. PPP3CC contains four regions, the catalytic domain, a regulatory subunit binding segment, a CAM-binding segment and an autoinhibitory helix [64,65]. As with the other PPs, the active site of PP2B contains two metal ions, Fe and Zn, and six amino acid residues (three His, two Asn, and one Asp) [63,64].

Upon the increase in intracellular Ca^2+^, CAM binds to the PPP3CC subunit, through the CAM binding region, causing the activation of the enzyme, by allowing it to access its substrates [19,63]. Since it was shown in 1990 that immature caput spermatozoa present higher levels of Ca^2+^ when compared to caudal sperm, a decline in PP2B activity throughout the epididymis journey was expected and further verified by Dey et al. in 2019 [35]. Furthermore, it was demonstrated that PP2B increases its activity during capacitation [35,66]. PP2B regulates GSK3 phosphorylation, preferentially dephosphorylating the GSK3α isoform, contrarily to PP1γ2 and PP2A which target both isoforms [19,35].

### 3.4. PP1γ2, PP2A and PP2B Interplay in the Regulation of Sperm Motility

Taken together, the data exposed in the previous section postulate that PP1γ2, PP2A and PP2B are strongly involved in spermatozoa motility regulation. In the last decades, the signaling pathways in which these PPs are involved have been deeply investigated (Figure 2). Indeed, apart from their individual role, an interplay between the three PPs has been proposed [2,4,19]. Notably, these PPs play distinct roles in motility at the different stages of spermatozoa maturation throughout the epididymis, as well as in hyperactivated motility. PP1γ2, PP2A and PP2B were collectively found to have consistently higher activity in the caput region (Figure 2a), where spermatozoa are immature and immotile, whereas they demonstrated low catalytic activity levels in the cauda (Figure 2b), where spermatozoa have acquired activated motility [4,19]. This suggests that the decrease of their catalytic activity is a requirement for motility acquisition and, when these PPs remain activated in caudal spermatozoa, motility acquisition is not achieved [7,8]. Concerning hyperactivated motility (Figure 2c), PP1γ2 and PP2A were found still catalytically inactive, while, paradoxically, PP2B presented phosphatase activity [19,35].

Considering the current state of the art, an interplay between both the PPs (PP1γ2, PP2A and PP2B) and PKs (GSK3 and PKA) can be proposed (Figure 2). In caput spermatozoa (Figure 2a), the increase in intracellular Ca^2+^ activates PP2B by promoting its interaction with the Ca^2+^-CAM complex. Consequently, PP2B preferentially dephosphorylates GSK3α at its inhibitory residue Ser21, activating it [35]. Synergistically, at this stage active PP2A is demethylated and phosphorylated, therefore being able to also dephosphorylate both GSK3 isoforms [57]. The PK that phosphorylates PP2A is still unclear. Active GSK3 phosphorylates the PP1γ2 inhibitor I2 at Thr73, which results in active PP1γ2, in a complex with only I3, since SDS22 is bound to p17 [51,57]. PP1γ2 not only contributes to GSK3 dephosphorylation, but also to key Ser and Thr residues dephosphorylation [4,57]. In caput spermatozoa, PKA presents no significant catalytic activity (recently reviewed by Dey et al., 2019 [19]).

During the spermatozoa’s journey through the epididymis (Figure 2b), the Ca^2+^ influx decreases, along with PP2B activity, rendering it inactive at caudal level and therefore unable to dephosphorylate the GSK3α isoform [35]. GSK3 dephosphorylation is further affected by the decrease in protein phosphatase methylesterase 1 (PPME1) activity in caudal spermatozoa, which increases PP2A methylation causing its inactivation. Highly phosphorylated GSK3 at Ser residues is inactive and incapable of phosphorylating I2, which can inhibit PP1γ2 [57]. Simultaneously, SDS22 is free from its interaction with p17, being in a complex with PP1γ2 as well [51]. The decrease in the activity of the Ser/Thr PPs, which causes a notable decrease in dephosphorylation, and an increase in the number of phosphorylated residues due to PKA activity, was observed in caudal spermatozoa [1,19,67]. The sAC, whose activity is regulated by HCO_3_^−^ and Ca^2+^ concentration, is activated and produces cAMP, activating PKA [67]. The concentration of cAMP in spermatozoa is also regulated by phosphodiesterases (PDEs) that can degrade it, being the equilibrium of sAC and PDE activity responsible for cAMP levels in spermatozoa [19]. Overall, active PKA in caudal spermatozoa not only phosphorylates both GSK3 isoforms (Ser21/9), but also other proteins, which seems to be a requirement for motility acquisition in the mature spermatozoon [2,19,50]. PP1γ2 was shown to be phosphorylated in caudal spermatozoa at its Thr320. The underlining mechanism is still unknown, but it is speculated that a cyclin-dependent kinase (CDK) is responsible for this residue’s phosphorylation [7,19,68].

At the FRT, hyperactivated motility is required for successful fertilization and both PP and PK play important roles [23,35] (Figure 2c). Ca^2+^ influx increases, again inducing PP2B activity, which in turn dephosphorylates GSK3α, being both enzymes active during capacitation [35]. GSK3α is now able to phosphorylate I2 which disassociates from PP1γ2. The other two inhibitors remain in a complex with this PP, rendering it still inactive. PP2A also remains inactive since it is methylated [4]. It was proposed by Battistone and colleagues that PPs downregulation during capacitation is also mediated by a Src family kinase (SFK). They showed that spermatozoon capacitating in the presence of a SFK inhibitor (SKI606) presented a decrease in phosphorylation levels, which was overcome by exposure to a PP inhibitor (OA). In addition, incubation with SKI606 also affected motility parameters which were similar to those of non-capacitated spermatozoa [31]. Concomitantly, the high concentrations of Ca^2+^ and HCO_3_^−^ stimulate cAMP production and a subsequent increase in PKA activity, which increases phosphorylation in Tyr residues among several known substrates within the spermatozoon, that seem to be required to achieve hyperactivated motility [31,69,70]. Remarkably, both PP2B and the GSK3α isoform present increased activity in hyperactivated spermatozoa, similar to their activity in caput immotile spermatozoa. Although many authors have been suggesting that the decrease in PP catalytic activity and concomitant increase in PK is a requirement for sperm motility, more recently, PP2B catalytic activity appeared to be essential for successful hyperactivation, accordingly to Dey et al. [31,35,36]. Comparing to activated progressive motility, the mechanisms that underline hyperactivated motility acquisition are more complex and are affected by alterations other than protein phosphorylation, which could explain the disparity between the PPs activity during this process [2,19,31,35,36,71].

Taking everything into account, the crosstalk between PP1γ2, PP2A and PP2B, as well as GSK3 and PKA, is essential both during spermatozoa maturation along the epididymis and capacitation at the FRT, since they determine the phosphorylation status of spermatozoa proteins, which appears to be crucial to initiate and maintain activated and hyperactivated motility. Regardless of the countless studies made to understand the biochemical mechanisms underlining sperm motility, several inconsistencies are yet to be solved and many protein interactions unveiled. For instance, some studies disagree on the state of PP2B activity during capacitation. Signorelli and colleagues verified its inactivity at this stage, while Dey et al. verified that PP2B presented catalytic activity during capacitation [35,36]. Furthermore, the PK and mechanisms that phosphorylates PP1γ2 at Thr320 are still unclear [7,19,68].

## 4. PP1, PP2A and PP2B Inhibition in Spermatozoa

PP activity modulation has already been accomplished and several studies showed that both natural and chemical compounds can be used to manage many diseases. The identification of naturally occurring small toxins capable of specifically inhibit PPs, largely contributed to the comprehension of Ser/Thr PPs role in various cellular events and other phosphorylation-dependent processes [12,13,14]. In fact, PPs are some of the most catalytically efficient enzymes, as they contain highly conserved active sites and do not possess high substrate specificity, which makes them very susceptible to inhibition by natural toxins [72,73]. Although most PP inhibitors present distinct chemical identities, they usually interact with a similar set of amino acids, along with the two metal ions they coordinate, that collectively compose the PP’s active site [12,74]. On the contrary, the different sensitivities that PPs present towards the inhibitors may be due to their very specific structural differences within the similarly folded catalytic core, despite their high degree of active site conservation [12]. After the discovery of OA [8,75], the first compound that was found to be a potent inhibitor of both PP1 and PP2A, many others emerged such as fostriecin, [76,77] cantharidic acid [8,78], Ciclosporin A (CsA) [79], cantharidin (CAN) and its analogues [80], cypermethrin (CYP) [81], tautomycin (TAU) [82], noludarin [83] and microcystins [83]. Except for CAN and fostriecin, most of these small molecules are too toxic for clinical use. Only CAN and its analogues have been developed showing antitumor activity and PP1 inhibition with lower cytotoxicity. [84,85] However, the use of most of those natural compounds for systemic use seems unlikely. It would be preferable to interfere with the interaction between the targeting subunit and the catalytic subunit, the binding of the targeting subunit to the target or the interaction of the targeting subunit with a regulator [86]. One approach to inhibit PP1 activity was through regulatory site targeting. Ammosova et al. demonstrated the viability of this approach by inhibiting the HIV-1 transcription and replication [87]. The authors employed small molecules targeting the RVxF-binding site of PP1 and identified a small molecule—1H4 (patent US 20090264463 A1)—that presumably binds to the RVxF motif of PP1. This molecule selectively disrupts the Tat-PP1 interaction without showing cytotoxicity and without affecting other PP1 holoenzymes [87]. Later, the same authors reported another molecule derived from 1H4 (1E7-03—patent CA 2881967 A1) which effectively inhibited HIV-1 transcription and suppressed replication of EBOV particles with high inhibitory properties and low cytotoxicity [88,89].

Inhibition of PP1, PP2A and PP2B in sperm samples was accomplished many times in the last century, which largely contributed to the comprehension of the signaling pathways discussed in the previous section (Table 1). Among the PP inhibitors discovered, only a few were ever tested in spermatozoa, namely CA, OA, DEL, CsA and E, which will be briefly described in this section.

### 4.1. Calyculin A (CA)

CA is an octamethylpolyhydroxylated C28 fatty acid, phosphoesterified, and linked to two γ-amino acids, which was first isolated from the marine sponge Discodermia calyx by Kato and colleagues in 1986 [15]. This toxin is cell membrane permeable and inhibits both PP1 and PP2A with IC50 values of 2 nM and 0.5 to 1.0 nM, respectively [12,78,103]. According to mutational analysis, CA might inhibit PP1 by interacting with its Tyr-272 [104].

CA was used several times for PP inhibition in spermatozoa from both animals and humans (Table 1). In 1994 and 1995, Ashizawa and colleagues first documented activation of motility following CA incubation in fowl spermatozoa, suggesting that inhibition of PP1 and PP2A stimulates motility [90,91]. Later, several authors performed experiments which reached a general similar outcome, an increase in sperm motility [8,9,36,57,92,98,99]. Goto et al. also demonstrated an increase in PKA phosphorylation, which caused its inactivation, as well as an increase in hyperactivated spermatozoa [93]. In 2021, Harayama et al. also demonstrated an increase in hyperactivation in boar spermatozoa [96]. Altogether, these studies highlighted PP1 and PP2A role in motility modulation, particularly hyperactivated motility.

### 4.2. Okadaic Acid (OA)

OA was first isolated from a marine sponge (*Halichondria okadai*), but later it was discovered that it was produced by a marine dinoflagellate (*Prorocentrum lima*) [16,105]. OA is a C-38 structure polyether with a C-38 structure that presents tumor promotion activity. It was reported to differentially inhibit PP1, PP2A and PP2B with IC_50_ values of 15–20, 0.1 and 3.6 nM, respectively [75,105]. Contrarily to CA, it has poor cell permeability, and these two inhibitors appear to compete for the same inhibitory site on PP2A [12,103]. OA and CA are frequently reported to be used in association, since they inhibit differentially the same PPs and attain similar results with respect to motility activation [8,9,94]. Nevertheless, Ashizawa and colleagues found that OA caused a less vigorous motility activation in fowl spermatozoa [92]. Ahmad et al. also verified changes in velocity along the curvilinear path (VCL) and amplitude of lateral head displacement (ALH) [61]. More recently, Dudiki et al. verified an increase in dimethyl PP2A and tyrosine phosphorylated PP2A [57]. Studies with this inhibitor reinforced PP1 and PP2A role in sperm motility and raised some questions about their differential functions.

### 4.3. Cyclosporin A (CsA)

Cyclosporin A (CsA) was isolated from the fungus *Tolypocladium inflatum* in 1992 and it is currently used as an immunosuppressive drug [12,79]. CsA inhibits PP2B with an IC_50_ of 5 nM. This inhibition is indirect since it first binds to cyclophilin and then it is able to interact with the patch region of PP2B. This inhibitor is also cell membrane permeant; however, it is difficult to handle in a laboratory due to its poor aqueous solubility, which explains why it is not used frequently [79,106,107]. Bennet and colleagues incubated CsA with human spermatozoa in a higher concentration than the one required to inhibit PP2B and speculated on the role of this PP in the process of acrosome reaction since its inhibition blocked its initial steps [101].

### 4.4. Deltamethrin (DEL)

Deltamethrin (DEL) is a type 2 synthetic pyrethroid currently used as an insecticide in agriculture and in the control of human disease vectors [81,108]. In fact, its use was recommended by the malaria control program of World Health Organization [108,109]. DEL is a very strong PP2B inhibitor, since its IC_50_ for this PP is around 100 pM [81]. Carrera and colleagues verified that DEL inhibited the Ca^2+^-stimulated dephosphorylation of human spermatozoa proteins and hypothesized that this dephosphorylation was mediated by a sperm-specific PP2B, rather than the known calcineurin [100]. In 2004, Ashizawa et al. determined that DEL was not able to restore motility at fowl body temperature, contrary to OA and CA [90,91,102]. Moreover, DEL significantly stimulated acrosome reaction [102]. The studies of Ashizawa and colleagues highlighted the differential role that PP2B plays in fowl spermatozoa motility when compared to PP1 and PP2A [90,91,102]. Signorelli et al. documented an increase in hyperactivated spermatozoa following DEL incubation, which suggested that PP2B activity downregulation is a requirement to achieve capacitation [36]. However, later this assumption was contradicted by Dudiki et al. [35]. This disparity might be due to the lack of consensus regarding how to objectively define and evaluate the hyperactivated motility pattern. In general, it is considered an increase in ALH and VCL, along with a decrease in the linearity of the trajectory (LIN); however, the cut-points are not well defined and differ between studies [110,111]. Indeed, both studies evaluated hyperactivation with different criteria, which affects their comparability. Dey et al. considered that a substantial increase in VAP (>100 µm/s) is indicative of hyperactivated motility alongside an increased VCL (>150 µm/s) and increased ALH, whereas Signorelli et al. relied on visual classification of hyperactivated spermatozoa, rather than the kinematic parameters of motility [35,36].

### 4.5. Endothall (E)

Endothall (E) is a synthetic herbicide, that constitutes a structural analogue of the PP inhibitor CAN [80]. It distinctively inhibits PP1 and PP2A with IC_50_ values of 5.0 μM and 90 nM respectively [112]. Signorelli et al. used E at a concentration that inhibits PP2A aiming to assess this PP’s role in the capacitation process. This incubation resulted in an increase in hyperactivated cells [36]. The choice of a distinct and less conventionally used PP inhibitor in this study allowed corroboration of previous findings, regarding PP activity with either CA or OA [36]. It would be interesting to compare the effects of endothall in spermatozoa with those of CAN and even other analogues, however, currently studies with this inhibitor are not reported.

## 5. Concluding Remarks

Many advances have been made concerning the understanding of spermatozoa motility acquisition and regulation. Notwithstanding, since the PP interplay is more complex than previously thought, many specific interactions and some inconsistencies are yet to be elucidated. Particularly, the state of activity of PP2B during hyperactivation, as well as the PK responsible to phosphorylate PP2A in caput epididymis spermatozoa and PP1 at both caudal motile and hyperactivated spermatozoa.

The discovery and use of PP inhibitors significantly contributed to the great advance of PP studies. However, to date, only a small portion of the inhibitors identified was reported to be tested in spermatozoa. Thus, additional studies with various concentrations of other PP inhibitors might bring novel insights into the intricate PP interplay. Some inhibitors constitute promising choices, particularly CAN, TAU and CYP. CAN and TAU inhibit both PP1 and PP2A, the first being a more potent PP2A inhibitor and the latter the strongest PP1 inhibitor identified so far. These two inhibitors were previously used in the clinical field, since CAN presents anticancer activity and TAU anticancer and immunosuppressor activity [12,82,113]. The use of both inhibitors in the same study allows a comparison of PP1 and PP2A activities and respective motility pattern alterations since, depending on the concentrations used, we could be inhibiting both PPs simultaneously or one at a time. Formerly, Suzuki et al. verified an increase in hyperactivated hamster spermatozoa after incubation with TAU [114]. Further studies with TAU could help analyze more deeply the motility pattern alterations assessed and explore other effects. CYP is a strong PP2B inhibitor, which was previously used solely to assess its toxic effects in human and rat spermatozoa, since it is a common pesticide. It was tested in concentrations highly superior to those required to inhibit PP2B, thereby, incubation with CYP, at concentrations closer to PP2B IC_50_, could help to assess if its activity is required to achieve the hyperactivated motility pattern or if this PP is inactive, as with PP1 and PP2A [81,115,116].

Indeed, since male infertility cases reach increasingly alarming numbers, the need of further investigation on this topic persists. By continuing to test the effects of compounds such as the PP inhibitors, we can not only contribute to the knowledge of the phosphorylation-dependent pathways underlining sperm motility, but also find solutions that could be applied in the clinical field to treat sperm motility-related conditions.

## Figures and Tables

**Figure 1 ijms-23-15235-f001:**
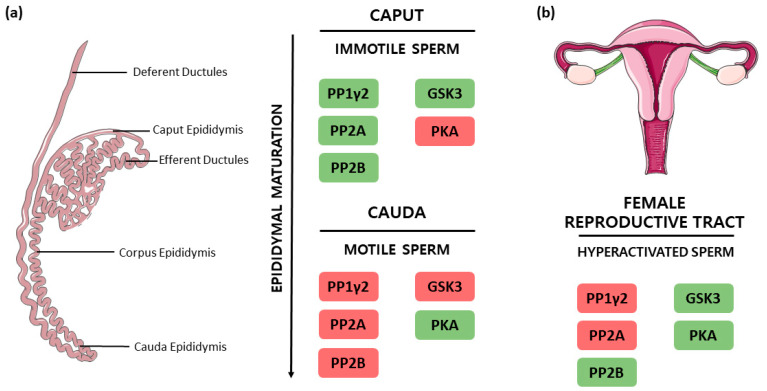
Representation of the PPs and PKs involved in activated motility acquisition in the epididymis, as well as in hyperactivation in the female reproductive tract. The colour green represents catalytic activity, whereas red stands for enzymatic inactivity. (**a**) Epididymis representation including its three epididymal subdivisions: caput, corpus, and cauda, as well as deferent and efferent ducts. In caput epididymis spermatozoa are immotile, the PPs PP1γ2, PP2A, PP2B and the PK GSK3 present catalytic activity, while PKA is inactive. In cauda, mature and progressively motile sperm are characterized by inactive PP1γ2, PP2A, PP2B and GSK3 and active PKA. (**b**) Feminine reproductive tract representation where hyperactivated spermatozoa presents inactive PP1γ2 and PP2A, whereas PP2B, GSK3 and PKA present catalytic activity.

**Figure 2 ijms-23-15235-f002:**
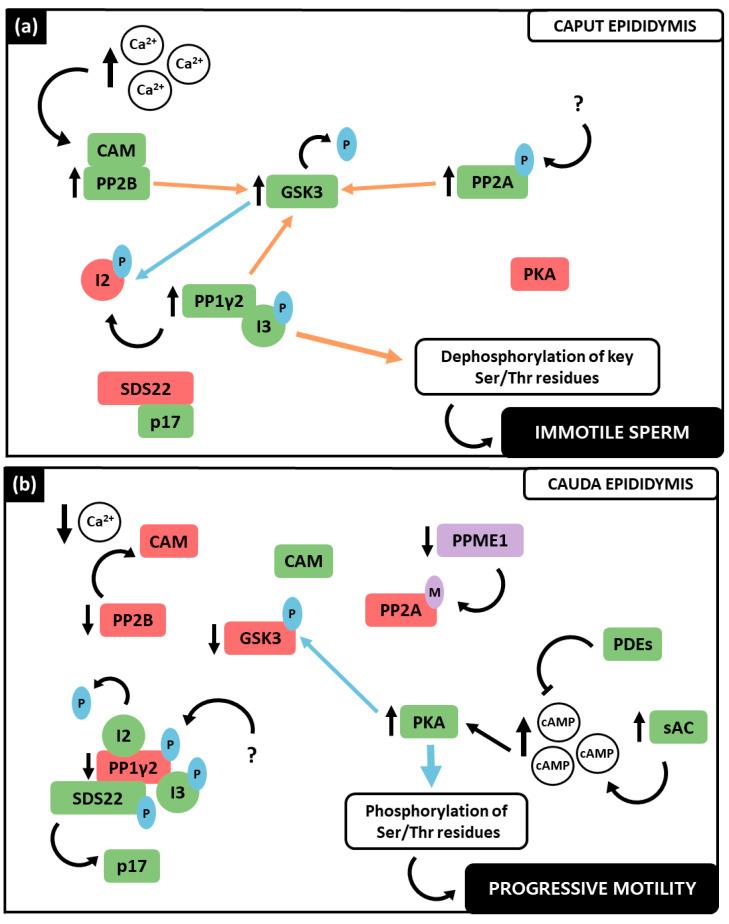
Interplay between the PPs, PP1γ2, PP2A and PP2B, and PKs, GSK3 and PKA, regarding sperm motility regulation. The color green represents catalytic activity, whereas red stands for enzymatic inactivity. Blue arrows define phosphorylation reactions, while orange represents dephosphorylation processes. (**a**) In caput immotile sperm, Ca^2+^ increase promotes PP2B interaction with Ca^2+^-CAM complex that activates it. PP2B dephosphorylates GSK3α increasing its activity. Phosphorylated PP2A also contributes to the phosphorylation of both GSK3 isoforms. GSK3 phosphorylates the I2 that disassociates from PP1γ2, rendering it active, solely in a complex with I3. The inhibitor SDS22 is bound to p17. PP1γ2 dephosphorylates GSK3 and Ser/Thr residues, resulting in immotile spermatozoa. PKA presents no significant catalytic activity. (**b**) In cauda epididymis, Ca^2+^ concentration is lower, causing CAM dissociation from PP2B and its subsequent inactivity. PPME1 decreases allowing PP2A methylation and inactivity. Simultaneously, sAC is activated and produces cAMP, which in turn activates PKA. cAMP degradation is due to PDE activity. PKA phosphorylates PP1γ2 and GSK3 that decreases its activity and is no longer able to phosphorylate I2, which forms a complex with PP1γ2 and SDS22 that dissociates from p17. Lastly, PKA’s increased activity causes phosphorylation of Ser/Thr residues which are a requirement for activated motility acquisition. (**c**) The increase in Ca^2+^ once again activates PP2B that dephosphorylates GSK3α rendering it active and able to phosphorylate I2. Due to sAC activation by HCO_3_^−^ and Ca^2+^, cAMP further increases and stimulates PKA activity. PKA phosphorylates GSK3. PP1y2 is phosphorylated and in a complex with SDS22 and I3. SFK also contributes to decrease PP1y2 activity along with PP2A, that remains methylated. The increase in phosphorylation of Tyr residues contributes to achieve hyperactivated motility.

**Table 1 ijms-23-15235-t001:** Reports on the inhibition of PP1, PP2A and PP2B by the PP inhibitors CA, OA, DEL, CsA and E in sperm. For each report, the PP inhibitor, spermatozoa model, concentration required for inhibition and outcome achieved is presented.

PP Inhibitor	Model	Concentration	Outcome	Reference
**Calyculin A**	Fowl sperm	0.1μM	Loss of motility following the addition of CaCl_2_ to demembranated spermatozoa, which was gradually restored by addition of EGTA.	Ashizawa et al. [90] 1994
		1.0 μM	Activation of intact sperm motility and stimulation of metabolic activity at 40 degrees.	Ashizawa et al. [91] 1995
		(1) Maximal effect: 1000 nM (2) 100 nM	(1) Induction of vigorous motility, stimulation of acrosome reaction in the presence of IPVL; (2) Significantly decreased ATP concentrations of spermatozoa.	Ashizawa et al. [92] 2006
	Mouse sperm	Maximal effects: 125 nM	Induced phosphorylation of several flagellar proteins, as well as PKA, inactivating it; Reduced progressive flagellar movement, inducing the hyperactivation-like motility pattern type.	Goto et al. [93] 2009
		0.1, 1, 3, 10, 100, 1000 nM	Overcome the block of capacitation-associated parameters by SKI606 and SU6656, such as PKA inhibition and tyrosine phosphorylation in a dose-dependent manner.	Krapf et al. [94] 2010
	Boar sperm	Maximal effect: 10 nM	Increased hypotonic volume, blocked the regulatory volume decrease (RVD) process, and increased relative cell volume.	Petrunkina et al. [95] 2007
		50 and 100 nM	Promotion of hyperactivation and cAMP-induced protein tyrosine phosphorylation identically at both concentrations.	Harayama et al. [96] 2012
	Bovine sperm	PP inhibition: 1.0 nM; Maximal effect: 3.4 nM	Activation of motility on caput and caudal spermatozoa; Demonstration of GSK3’s presence in bovine sperm.	Vijayaraghavan et al. [9] 1996
		50 nM	Increase of phosphorylated PP1γ2 in both caput and caudal epididymal spermatozoa.	Huang et al. [97] 2004
	Monkey sperm	0.59 nM	Increase in %motility and an acceleration in mean path velocity;	Smith et al. [8] 1996
		100 nM	Increase in motile cells of the caput sperm, without any effect on their path velocity.	Smith et al. [98] 1999
	Human sperm	IC_50_: 0.75 nM	Increase in %motility and an acceleration in mean path velocity; Demonstration that sperm contains PP1 and its regulators.	Smith et al. [8] 1996
		100 nM	Increase in p105/81 phosphotyrosine levels and stimulation of sperm capacitation.	Leclerc et al. [99] 1996
**Okadaic acid**	Fowl sperm	1.0 μM	Loss of motility following the addition of CaCl_2_ to demembranated sperm, which was gradually restored by addition of EGTA.	Ashizawa et al. [90] 1994
		Maximal effect: 1000 nM	Less vigorous motility stimulation, induction of acrosome reaction in the presence of IPVL.	Ashizawa et al. [92] 2006
	Mouse sperm	0.1, 1, 3, 10, 100, 1000 nM	Overcome the block of capacitation-associated parameters by SKI606 and SU6656, such as PKA inhibition and tyrosine phosphorylation in a dose-dependent manner.	Krapf et al. [94] 2010
	Boar sperm	Maximal effect: 10 nM	Increased hypotonic volume, blocked the regulatory volume decrease (RVD) process, and increased relative cell volume.	Petrunkina et al. [95] 2007
	Bovine sperm	PP inhibition: 1 μM; Maximal effect: 5 μM	Activation of motility on caput and caudal sperm; Demonstration of GSK3’s presence in bovine sperm.	Vijayaraghavan et al. [9] 1996
		5 nM	Increase of sperm motility parameters (%motility, velocity, and lateral head amplitude), as well as elevation of dimethyl PP2A and tyrosine phosphorylated PP2A.	Dudiki et al. [57] 2015
	Monkey sperm	37.2 nM	Increase in %motility and an acceleration in mean path velocity.	Smith et al. [8] 1996
	Human sperm	1.0 μM	Alteration of velocity along the curvilinear path and amplitude of the lateral displacement of the head; Inhibition of Ca^2+^-dependent dephosphorylation of cAMP-dependent phosphoproteins in capacitating sperm.	Ahmad et al. [61] 1995
		38.8 nM	Increase in %motility and an acceleration in mean path velocity; Demonstration that sperm contains PP1 and its regulators.	Smith et al. [8] 1996
		(1) 1 μM (2) 100 nM	(1) Increase in p105/81 phosphotyrosine levels; (2) Stimulation of sperm capacitation.	Leclerc et al. [99] 1996
		100 nM	Inhibited the Ca^2+^-stimulated dephosphorylation of human sperm phosphotyrosine-containing proteins.	Carrera et al. [100] 1996
		(IC_50_) PP1: 10 nM PP2A: 0.1 nM	Increased phosphorylation on threonine residues; Demonstration that the activity of this PP decreases during the capacitation process.	Signorelli et al. [36] 2013
**Cyclosporin A**	Human sperm	2 μM	Blocked acrosomal exocytosis, suggesting PP2B is required in the early steps of the secretory process of the acrosome reaction.	Bennet et al. [101] 2010
**Deltamethrin**	Fowl sperm	1–100 nM Maximal effect: 10 nM.	Did not permit the restoration of motility at 40 °C but stimulated the acrosome reaction in the presence of IPVL.	Ashizawa et al. [102] 2004
	Human sperm	10 nM;	Inhibited the Ca^2+^-stimulated dephosphorylation of human sperm phosphotyrosine-containing proteins.	Carrera et al. [100] 1996
		(IC_50_) PP2B: 0.1 nM	Increased phosphorylation on threonine residues; Demonstration that the activity of this PP decreases during the capacitation process.	Signorelli et al. [36] 2013
**Endothall**	Human sperm	(IC_50_) PP2A: 90 nM	Increased phosphorylation on threonine residues; Demonstration that the activity of this PP decreases during the capacitation process.	Signorelli et al. [36] 2013

## Data Availability

Data sharing is not applicable to this article as no new data were created or analysed in this study.

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
