# Peer review of "PP1, PP2A and PP2B Interplay in the Regulation of Sperm Motility: Lessons from Protein Phosphatase Inhibitors"

_ijms, 2022, doi:10.3390/ijms232315235_

Round 1
Reviewer 1 Report
Manuscript id: ijms-2019049
The review article by Ferreira A and colleagues covers the crucial role of regulatory protein players that play significant roles in spermatozoa motility and eventually egg fertilization. The review is focused on biochemical pathways involving specific protein phosphatases and the role of naturally occurring inhibitors that affect motility and hyperactivation of the spermatozoa. Overall, this is an interesting article that connects the missing links on several aspects of sperm maturation and motility. The article is well written and is supported by graphic figures. I have few suggestions that could help improve the manuscript before getting published as a biomolecule review article.
Comment 1: The introduction section needs to be refined for a consistent flow of information. Please elaborate on the information in line 42-45 and then correlate with the importance of inhibitors.
Comment 2: The authors are also suggested to add more information on the clinical and therapeutic aspects of sperm specific phosphatases as well as their inhibitors.
Comment 3: Several pharmacological compounds have been described that enhance/inhibit sperm motility and fertilizing capacity. Addition of a section describing small molecules/drugs that specifically affect the phosphatases and kinases will certainly enhance the review article.

Author Response
Manuscript ID: ijms-2019049
Title: PP1, PP2A and PP2B interplay in the regulation of sperm motility: lessons from protein phosphatase inhibitors
Reviewer #1
The review article by Ferreira A and colleagues covers the crucial role of regulatory protein players that play significant roles in spermatozoa motility and eventually egg fertilization. The review is focused on biochemical pathways involving specific protein phosphatases and the role of naturally occurring inhibitors that affect motility and hyperactivation of the spermatozoa. Overall, this is an interesting article that connects the missing links on several aspects of sperm maturation and motility. The article is well written and is supported by graphic figures. I have few suggestions that could help improve the manuscript before getting published as a biomolecule review article.
Reply: Thank you for your kind comments and suggestions which we believe will improve the manuscript. Your suggestions were carefully considered and incorporated throughout the manuscript (track changes).
Comment 1: The introduction section needs to be refined for a consistent flow of information. Please elaborate on the information in line 42-45 and then correlate with the importance of inhibitors.
Reply: The introduction was reorganized, and we elaborated a little more about the importance of the use of PP inhibitors, as suggested.
Comment 2: The authors are also suggested to add more information on the clinical and therapeutic aspects of sperm specific phosphatases as well as their inhibitors.
Reply: Thank you for your suggestion. The following information was added to page 8, lines 352-355: “Except for cantharidin and fostriecin these small molecules are too toxic for clinical use, thus, only cantharidin and cantharidin analogues have been developed showing antitumor activity and PP1 inhibition with lower cytotoxicity. However, the use of most of those natural compounds for systemic use seems unlikely.”.
Comment 3: Several pharmacological compounds have been described that enhance/inhibit sperm motility and fertilizing capacity. Addition of a section describing small molecules/drugs that specifically affect the phosphatases and kinases will certainly enhance the review article.
Reply: We appreciate your suggestion. Considering that the main goal of this manuscript is to discuss the use of PP inhibitors to unveil the molecular mechanisms involved in sperm motility acquisition, we do not consider relevant to include other pharmacological compounds that modulate sperm motility without affecting the phosphatase proteins themselves or with unknown mechanisms of action. Nevertheless, we include the information about other small molecules/drugs that specifically inhibit phosphatases, as suggested. This information was added (page 8).
For addittional information, please see the attachment

Reviewer 2 Report
Summary:
The present study reflects the authors' efforts to analyse the data of a number of publications related to the role of sperm-specific protein phosphatases in the pathogenesis of male infertility.
The subject matter is current and interesting, primarily because of the difficulties in the treatment of the disorder.
The authors presented their results in a logical and meaningful manner, using tables and figures for the better interpretation of the data. The conclusions of the work are clear and are well supported by the results. The study will serve as a helpful document in this field of research.
Nevertheless, some minor revisions are recommended before publication.
Observations:
Line 78: “as” should be “such as”.
Line 203: “being PPP3CC the sperm-specific” should be “PPP3CCC being the sperm-specific”.
Line 205: “being the latest present” should be “the latest being present”.
Line 299: there is no need for a comma after “(SKI606)”.
Line 445: there is no need for a comma after “PP inhibitors”

Author Response
Manuscript ID: ijms-2019049
Title: PP1, PP2A and PP2B interplay in the regulation of sperm motility: lessons from protein phosphatase inhibitors
Summary:
The present study reflects the authors' efforts to analyse the data of a number of publications related to the role of sperm-specific protein phosphatases in the pathogenesis of male infertility.
The subject matter is current and interesting, primarily because of the difficulties in the treatment of the disorder.
The authors presented their results in a logical and meaningful manner, using tables and figures for the better interpretation of the data. The conclusions of the work are clear and are well supported by the results. The study will serve as a helpful document in this field of research.
Nevertheless, some minor revisions are recommended before publication.
Reply: We thank the Reviewer for the valuable comments and suggestions. All the minor points indicated by the reviewer were addressed (track-changes).
Observations:
Line 78: “as” should be “such as”.
Reply: The correction was made.
Line 203: “being PPP3CC the sperm-specific” should be “PPP3CCC being the sperm-specific”.
Reply: The sentence was corrected.
Line 205: “being the latest present” should be “the latest being present”.
Reply: Correction was made.
Line 299: there is no need for a comma after “(SKI606)”.
Reply: The comma was removed.
Line 445: there is no need for a comma after “PP inhibitors”
Reply: The comma was removed.
For addittional information, please see the attachment

Reviewer 3 Report
The authors' work shows a well thought-out research.
I recommend that the article be accepted without modification.
Author Response
Manuscript ID: ijms-2019049
Title: PP1, PP2A and PP2B interplay in the regulation of sperm motility: lessons from protein phosphatase inhibitors
The authors' work shows a well thought-out research.
I recommend that the article be accepted without modification.
Reply: Thank you for taking the time to review our article and for your kind comments. We appreciate your support and acknowledge your recommendation to publish our work. Concerning the English language and style, the reviewer indicated “English very difficult to understand/incomprehensible”; however, he/she also gave 4* to the question “Is the English used correct and readable?”. So, we are not sure whether the English needs extensive editing or not. Nevertheless, we carefully revised the entire manuscript, improving the language used and the clarity of the speech, as suggested.
